# Understanding the Biological Relationship between Migraine and Depression

**DOI:** 10.3390/biom14020163

**Published:** 2024-01-30

**Authors:** Adrián Viudez-Martínez, Abraham B. Torregrosa, Francisco Navarrete, María Salud García-Gutiérrez

**Affiliations:** 1Hospital Pharmacy Service, Hospital General Dr. Balmis de Alicante, 03010 Alicante, Spain; aviudezmartinez@gmail.com; 2Instituto de Neurociencias, Universidad Miguel Hernández, 03550 San Juan de Alicante, Spain; a.bailen@umh.es (A.B.T.); fnavarrete@umh.es (F.N.); 3Research Network on Primary Addictions, Instituto de Salud Carlos III, MICINN and FEDER, 28029 Madrid, Spain; 4Instituto de Investigación Sanitaria y Biomédica de Alicante (ISABIAL), 03010 Alicante, Spain

**Keywords:** migraine, major depressive disorder, serotonin, neuropeptides, sexual hormones, immune system

## Abstract

Migraine is a highly prevalent neurological disorder. Among the risk factors identified, psychiatric comorbidities, such as depression, seem to play an important role in its onset and clinical course. Patients with migraine are 2.5 times more likely to develop a depressive disorder; this risk becomes even higher in patients suffering from chronic migraine or migraine with aura. This relationship is bidirectional, since depression also predicts an earlier/worse onset of migraine, increasing the risk of migraine chronicity and, consequently, requiring a higher healthcare expenditure compared to migraine alone. All these data suggest that migraine and depression may share overlapping biological mechanisms. Herein, this review explores this topic in further detail: firstly, by introducing the common epidemiological and risk factors for this comorbidity; secondly, by focusing on providing the cumulative evidence of common biological aspects, with a particular emphasis on the serotoninergic system, neuropeptides such as calcitonin-gene-related peptide (CGRP), pituitary adenylate cyclase-activating polypeptide (PACAP), substance P, neuropeptide Y and orexins, sexual hormones, and the immune system; lastly, by remarking on the future challenges required to elucidate the etiopathological mechanisms of migraine and depression and providing updated information regarding new key targets for the pharmacological treatment of these clinical entities.

## 1. Introduction

Migraine is a complex neurological disorder affecting more than 10% of the general population, causing a marked loss of productivity and quality of life. In fact, it is placed as the sixth most disabling disease worldwide. Its clinical features include intense and pulsating head pain localized unilaterally with a variable duration of up to 72 h that may/may not be associated with aura, a sequence of visual or sensory disturbances, such as flashing lights and haphazard lines, that occur shortly before a migraine attack. Additionally, these symptoms may be accompanied by nausea, vomiting, and hypersensitivity to acoustic, olfactory, or visual stimuli. Several etiological factors seem to play a part in the development of migraine, including genetic background, climatic region, socioeconomic status, and lifestyle. Genetic, sex-related, and environmental factors seem to contribute the most, with women being approximately three times more likely than men to develop it and with up to 12% of Caucasians suffering from this neurological condition [1].

The International Headache Society classifies migraine into episodic migraine or chronic migraine, depending on the frequency of outbursts. To add more complexity, cross-sectional and longitudinal studies indicate a close relationship between migraine and mood disorders, especially depression, which has been identified as an independent risk factor for developing migraine earlier and with a worse prognosis [2,3,4]. This relationship has been proposed to be bidirectional, with migraine also increasing the risk of developing depression by 2.5 times [5,6,7]. Therefore, the comorbidity of migraine and depression is frequent and, foremost, hinders the diagnosis and treatment of migraine by increasing the total costs and disability associated with both disorders and diminishing patients’ quality of life.

Despite this association being attributed to shared genetic factors—an increased risk of depression in siblings and twins affected by migraine has been demonstrated [8,9,10]—it has also been proposed that both entities may share neurotransmission pathways and neurobiological features; yet, their pathophysiology is complex and has not been fully characterized. Until a few years ago, depression was mostly believed to involve central mechanisms and migraine to involve peripheral alterations, such as sensitized perivascular trigeminal nociceptors [11]; however, in recent studies, migraineurs were also found to feature abnormal cortical sensory processing [12] and altered central modulation [13]. Thus, neurotransmitters such as serotonin, hormones, and neuropeptides, and even genetic and functional brain alterations have been linked with migraine. The immune system seems to partially explain the etiology of both disorders. Therefore, this review aims to explore the biological relationship between migraine and depression. The identification of the mechanisms involved in the occurrence of depression in migraineurs could help identify key targets that may serve as biomarkers for preventing this comorbidity or develop more effective pharmacotherapeutic approaches that, ultimately, would help to improve patient quality of life and the clinical strategies addressed to treat these neuropsychiatric conditions.

## 2. Materials and Methods

The literature review consisted of a search for scientific information in the Medline database (PubMed) employing Medical Subject Headings (MeSH) related to the topic of the review: “migraine”, “depressive disorder”, and “mechanisms”. These terms were combined by the Boolean operator “AND”. Moreover, to maximize the selection of information, additional search equations were used employing the MeSH “migraine” AND “system alterations” and “migraine” AND “depressive disorder”.

All the authors critically analyzed all the results for each search to decide upon the selection of each reference according to the alignment of its content with the subject matter of the study. No PubMed filters were applied to maximize the selection of all the available and appropriate information. All original articles, systematic reviews, or meta-analyses focusing on migraine and depression were accepted. Those articles unrelated to the topic of interest, not written in English, or to which access was impossible were excluded.

## 3. Results

### 3.1. Serotonin

While sufficient evidence has not been provided for any single system/endogenous substance to explain the bidirectional connection between migraine and depression on its own, a dysfunction in the serotoninergic system has been continuously considered one of the major contributors. Brain serotonin imbalance seems to be implicated in both diseases. Additionally, pharmacological modulation of the serotoninergic system is one of the main targets in both disorders. In this section, we highlight the findings accumulated over the years supporting the involvement of the serotonergic system in this comorbidity (Figure 1).

The role of serotonin (5-HT) in the pathophysiology of migraine has been a matter of study for decades. The first evidence was provided in the early 1960s by Sicuteri et al. [14]. In this study, increased levels of the 5-hydroxyindoleacetic acid, the main metabolite of 5-HT, were found in the urine of patients during migraine attacks. This elevation is representative of higher plasma levels of serotonin. Subsequent investigations found that plasma levels of 5-HT decrease between attacks and increase during attacks [15,16]. These findings laid the foundations for the theory that postulates that migraine is a syndrome of chronically low interictal levels of 5-HT with a transient increase during attacks [17,18].

Considering that 5-HT is a crucial neurotransmitter for the coherent modulation of peripheral and central pain signaling [19,20,21], the abnormalities observed in migraine patients have been considered as an indicator of pain-modulating system dysfunction. Moreover, 5-HT is involved in excitatory (hyperalgesia) as well as in inhibitory (analgesia) mechanisms [22]; yet, its role is determined by location, cell type, and serotonin receptor subtype, among other factors. Before and after migraine attacks, patients present an increased sensitivity to visual (light), auditory (sound), or somatic stimuli (allodynia). The investigations focused on this phenomenon have shown that patients present a ‘deficient habituation’ during the pain-free period. Interestingly, habituation changes with the proximity of an attack, during the attack, and the course of migraine (episodic or chronic), which is characterized by the occurrence of the sensitization phenomena [23]. The conversion from episodic to chronic migraine is not well understood; however, two putative mechanisms that seem to play a remarkable part are the increased excitability of neurons in central nociceptive pathways and a dysfunctional pain modulation, both regulated by 5-HT. In this sense, decreased serotonin availability has been associated with abnormal habituation in episodic migraine [24,25,26,27]. Moreover, it has been proposed that the low levels of 5-HT induce a disinhibition of pain signals from the peripheral nociceptor, resulting in a decrease of nociceptive thresholds and increased responsiveness to sensory or somatic stimuli, making patients more susceptible to stimuli [28,29,30,31,32,33,34,35]. Serotonin surges during migraine attacks would increase and maintain pain [19].

Solid evidence supporting the involvement of 5-HT in the pathophysiology of migraine is linked to the efficacy of 5-HT_2_ antagonists, such as methysergide and pizotifen, as prophylactic drugs and 5-HT_1B/D_ agonists, such as triptans, for acute management. Triptans, the pillar of migraine therapy, increase serotonin signaling in cranial blood vessels and nerve endings [36], relieving pain by inducing vasoconstriction and reducing the release of vasoactive peptides, such as the calcitonin-gene-related peptide (CGRP) and substance P (SP), among others [37,38].

The relevance of 5-HT in migraine has also been assessed by studying polymorphisms in the gene *SLC6A4*, coding for the serotonin transporter (SERT), involved in the removal of serotonin from the synaptic cleft back to the presynaptic neuron. Genetic alterations in SERT can partially explain changes in serotonin levels. More precisely, two polymorphisms in SERT have been linked with migraine, VNTR STin2 and 5-HTTLPR. The VNTR STin2 12/12 genotype is linked with migraine susceptibility in those of European descent [39] and in the general population [40]. In the case of the 5-HTTLPR polymorphism, the short variant (S) is associated with slower clearing of 5-HT from the synaptic cleft, increasing the risk of migraine development [41,42,43], having some negative results [44,45].

More recently, genetic studies have shown that migraine is a polygenetic disorder, since more than 38 loci have been linked to a higher susceptibility to migraine [46]. Locus 1p36 for the 5-HT_1D_ receptor and functional polymorphism rs3813929 of the promoter region of the gene for the 5-HT_2C_ receptor have also been associated with migraine [47,48]. Additionally, the T allele of this gene impacts the transcription rate of the 5-HT_2C_ receptor and was found in Turkish population with migraine [48].

Neuroimaging studies have also examined the relevance of 5-HT in migraine. PET studies have found increased 5-HT synthesis and elevated 5-HT turnover, which result in decreased brain 5-HT levels [49]. However, in other studies, no differences were observed regarding the rates of 5-HT synthesis [50]. Several variables appear to influence the discrepancies of these results, namely, the time elapsed since the last attack as well as clinical and demographic variables (age, sex, diagnosis, and chronicity). Further studies have found lower 5-HT_4_ receptor binding, which was interpreted as higher interictal brain 5-HT levels [51,52]. However, increased 5-HT_1A_ binding was found in the brainstem of patients during migraine attacks, a change that has been associated with decreased 5-HT availability [53]. Curiously, sumatriptan appears to reverse the increase in 5-HT synthesis in the brain during migraine attacks [27]. These results suggest that the brain’s serotonin synthesis rate may be altered in migraineurs and that triptans are effective in modulating pain pathways by decreasing brain serotonin synthesis.

The role of serotonin in depression is well documented, with low 5-HT levels being considered one of the most validated biological causes of mood disorders [54,55]. Thus, 5-HT levels in the cerebrospinal fluid are associated with the severity of depression in major depressive disorder (MDD) patients [56]. Moreover, as described in the case of migraine, the 5-HTTLPR polymorphism has also been associated with depression and the efficacy of antidepressants. Lee et al. found a positive correlation of this polymorphism with depression [57]. Accordingly, additional studies showed that this polymorphism displays a role in modulating the onset of mood disorders [6,58]. The s/s genotype is linked with worse antidepressant response, reduced serotonin expression and function, and increased fear and anxiety [59,60]. Additionally, depressed patients with l/l or l/s genotypes showed better responses to selective serotonin reuptake inhibitors (SSRIs) [61,62]. A few studies have analyzed the 5-HTTLPR polymorphism in migraine patients with depressive symptoms, observing no association with the onset of migraine combined with depression [63,64]. However, the sample size was small, making further large-scale studies necessary.

Patients affected by mood disorders also present alterations in 5-HT_1D_, the pharmacological target of triptans, including reduced sensitivity, density, and binding of central 5-HT_1D_ receptors [65,66]. Noticeably, the cessation of long-term excessive triptan use is associated with the onset of severe major depression [67]. Therefore, it is hypothesized that the chronic excessive use of triptans may induce persistent changes in the serotoninergic system, including the desensitization of 5-HT_1_ receptors [68].

Magnetic resonance imaging (MRI) showed that the comorbidity of migraine and depression is associated with a more pronounced reduction in brain volume [69]. This study provided the first evidence providing that migraineurs with depression may represent a clinical phenotype with different long-term sequelae. Additional regions affected by migraine–depression comorbidity are the thalamus and the fusiform gyrus [70]. Patients suffering from this comorbidity showed a marked decrease in the intrinsic brain activity in the thalamus [71]. Moreover, the activity of the medial prefrontal cortex is altered in both diseases, which can influence the activation of the dorsal raphe nucleus, leading to depressive symptoms and headaches [71,72]. By using transcranial sonography (TCS), a real-time imaging technique, reduced echogenicity has been identified in the midbrain raphe (MBR) of MDD patients [73]. This alteration is representative of structural changes in the MBR, which can explain the monoamine deficiency hypothesis in depression [74]. Interestingly, the reduced echogenicity in the MBR has also been associated with depressive symptoms, migraine attack frequency, or overuse of analgesics in migraine patients [75,76,77,78]. Interestingly, the microarchitecture of the cerebral cortex is different in patients with MDD and migraine compared to patients diagnosed with only one of them [70,79]. Altogether, this evidence firmly supports the involvement of 5-HT in migraine–depression comorbidity. Figure 1Representative figure of the involvement of the serotoninergic system in migraine and depression. TNC: trigeminal nuclear complex; 5-HT_1B_: serotonin receptor 1B; 5-HT_1D_: serotonin receptor 1D; 5-HT_1F_: serotonin receptor 1F; CGRP: calcitonin gene-related peptide; 5-HTTLPR polymorphism: functional polymorphism in serotonin transporter; mPFC: medial prefrontal cortex; MBR: midbrain raphe. Figure adapted from [80].
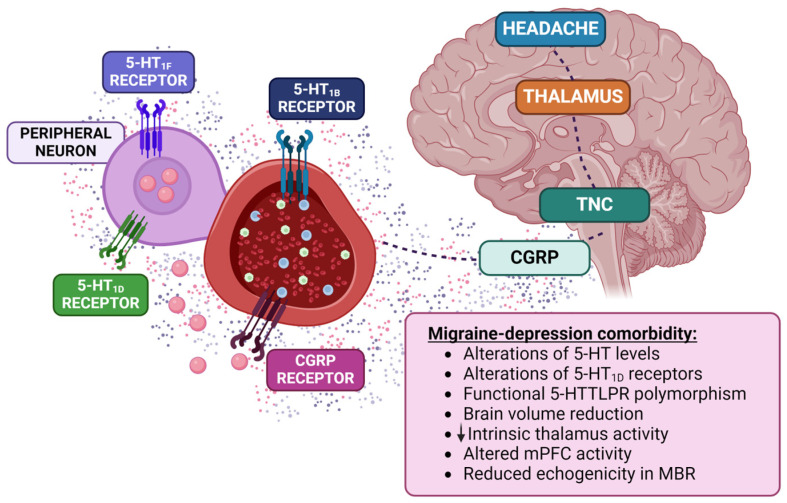


### 3.2. Neuropeptides

#### 3.2.1. Calcitonin-Gene-Related Peptide

Calcitonin gene-related peptide (CGRP) is a 37-amino acid, vasoactive, neuroendocrine peptide that belongs to the calcitonin family. Both of its isoforms (α, β) are closely related peptides encoded by tandem genes expressed in human chromosome 11 (CALCA and CALCB, respectively) and only differ in a few amino acids, dependent upon the species. Thus, though CGRPα and CGRPβ peptides are differentially regulated, they have nearly indistinguishable activities and are expressed in an overlapping pattern [81,82] in the nervous, cardiovascular, immune, hematopoietic, and gastrointestinal systems [81,83,84,85].

Regarding the nervous system, CGRP is expressed in the peripheral ganglia and the central nervous system (CNS) [85]. In the periphery, CGRP is released from primary afferents of the trigeminal nerve into the perivascular space of the meninges, as well as within the ganglia, inducing a crosstalk that involves CGRP, purinergic receptors, nitrous oxide (NO), and inflammatory cytokines [86]. This scenario generates a perfect storm combining vasodilatation and positive feedback loops, sensitizing trigeminal ganglia (TG) neurons, ultimately contributing to pain exacerbation and peripheral sensitization, as in migraine [81].

In fact, different animal models and clinical studies have established a direct correlation between an augmentation of CGRP blood concentrations and the onset/worsening of symptoms in migraine-like models and migraineurs [86]. Firstly, indirect approaches, such as electrical stimulation of the TG, the dural surface of the dura matter, or the administration of nitroglycerine, a precursor of NO, caused a noticeable release of CGRP, which in turn caused vasodilation, increased meningeal blood flow [87,88], and trigeminal system sensitization in animal models [89,90]. These results consolidate the crosstalk between the CGRP, NO, TG, and meninges stated above and are coherent with the increase in CGRP levels observed in patients with chronic migraine [91,92,93], though this feature is not found in all migraineurs.

Secondly, direct administration of CGRP has been proven to provoke peripheral and central sensitization, depending on the administration route selected [86]. Moreover, direct infusion of CGRP causes a dilation of the cortical pial arteries and arterioles and of the middle meningeal artery. This vasodilatation increases local cortical cerebral blood flow, provoking migraine-like symptoms in rodent models [91,92,93]. These effects have also been reported in patients. After intravenous infusion of CGRP at a dose able to induce vasodilation, 66% of migraineurs experienced a migraine-like headache [94,95,96]; yet, patients with no previous history of migraine only suffered mild headaches [97].

All the effects and symptoms expressed above are partially or totally abolished after the administration of indirect CGRP modulators that exert 5-HT_1B/1D_ receptors agonist properties in the serotoninergic pathway, such as triptans [98], and drugs that directly complex CGRP or block its receptors, such as gepants [99] and anti-CGRP monoclonal antibodies [100,101,102].

Regarding depression, CGRP disturbances have also been observed in several animal models. For example, Shao et al. reported increased CGRP levels in the CSF and hippocampus (HIPP) of rats, showing more prominent depressive-like behavior in a model of post-stroke depression [103]. CGRP could also play a part in depression onset, as shown by Jiao et al., who reported increased depressive-like behavior in mice after the central administration of CGRPα. Moreover, these findings were counteracted after the administration of CGRP antagonists [104].

These results are controversial, since other authors have found CGRP-overexpressing mice to show stress-resistant behavior [105,106] or normal sensitivity to stress [107]. Nevertheless, recent published work seems to confirm that increased brain levels of CGRP are present in well-established animal models of depression; yet, these are not modified by SSRIs or tricyclic antidepressants (TCAs) [108].

Accordingly, an increase in CGRP levels has been found in women with MDD compared to healthy subjects, suggesting a potential role for CGRP as a biomarker [109]. Some studies even suggested that the fold elevations of CGRP could be related to symptom severity rather than disease classification per se [110]. Interestingly, recently published results from a prospective cohort show how the treatment with some anti-CGRP monoclonal antibodies led to improvement in depressive symptoms in individuals with migraine, with independence of migraine reduction [111]. Altogether, these findings support the rationale behind the potential role of CGRP in both entities (Figure 2), migraine and depression; though further studies are required, this could provide an innovative approach for patients that could benefit from undergoing treatment targeting CGRP.

#### 3.2.2. Pituitary Adenylate Cyclase-Activating Polypeptide

Pituitary adenylate cyclase-activating polypeptide (PACAP), which is found as a 38-amino-acid peptide (PACAP-38) and its truncated 27-amino-acid form (PACAP-27), belongs to the vasoactive intestinal polypeptide (VIP)–secretin–growth hormone-releasing hormone–glucagon superfamily [113]. Both isoforms act on the same receptors and are encoded by exon 4 of the ADCYAP1 gene, located in chromosome 18, with PACAP27 being the only one that suffers post-translational shortening at the C-terminus site. Despite having similar affinities and functions, PACAP-38 is the predominant peptide, representing more than 90% of the total PACAP content in most tissues, including the CNS [112].

PACAP38 is also expressed in migraine-related anatomical structures located in the peripheral nervous system, such as the sphenopalatine ganglion, parasympathetic perivascular nerve fibers, and sensory nerve afferents of the cranial arteries. Despite being co-localized with CGRP-immunoreactive neurons in the TG and trigeminal nucleus caudalis, PACAP38 has a larger parasympathetic distribution and a smaller trigeminal distribution than GCRP [112,114,115]. Thus, it could be hypothesized that PACAP38 functions primarily as a neuropeptide in the parasympathetic pathways underlying migraine, while CGRP acts predominantly in the sensitization process [112].

In addition, the available evidence confirms that PACAP38 triggers a migraine-like response similar to CGRP administration in some animal models of migraine [116,117]. However, it is important to notice that the PACAP38 pathway seems to be distinct and independent from other migraine pathways [112,116,117], contrasting previous observations [118]. In this sense, multiple potential mechanisms have been proposed to explain the migraine-inducing effect of PACAP38, including: vasodilatation, modulation of the parasympathetic nervous system, mast cell degranulation, activation of sensory afferents, and central effects [112].

Several animal studies have demonstrated that PACAP38 can activate the trigeminovascular system, causing perivascular neurogenic inflammation mediated by mast cell degranulation [119,120]. Its peripheral injection has also been shown to induce hyperalgesia [121] and light aversive behavior in wild-type mice, which is absent in PACAP knockout mice [122]. Despite exerting vasodilatory effects [123], it has been suggested that the ability of PACAP-38 to induce migraine may not be related to this feature but to its excitatory role in pain transmission [124,125].

In humans, the administration of PACAP38 caused headaches in healthy volunteers [126,127], and plasma levels of PACAP38 seemed to be elevated during spontaneous migraine attacks compared to the interictal phase. However, blood levels decreased during the interictal phase compared to healthy volunteers, a phenomenon attributed to the chronic depletion of PACAP38 caused by an excessive consumption during migraine crisis [128,129]. A double-blind, randomized, placebo-controlled study found that intravenous administration of PACAP38, but not placebo, caused headache in healthy subjects and migraineurs without aura. Moreover, in 58% of migraineurs without aura, a migraine-like attack was observed, while only 16% of patients in the placebo-controlled group experienced migraine-like symptoms [130]. Therefore, it could be hypothesized that PACAP38 antagonists may be a therapeutic tool.

Nevertheless, in a phase 2a, randomized, double-blind, placebo-controlled, three-arm clinical trial aimed to evaluate the efficacy and safety of AMG 301 (a PACAP38 antagonist) in migraine prophylaxis, no statistically significant differences were found between AMG 301 and placebo [131]. Even though no favorable results were obtained, more promising results have been reported regarding ALD1910, a monoclonal antibody 4000-fold more selective for PACAP38 and PACAP27 than VIP, in an umbellulone-induced rat model of headache [132]. More recently, Lu AG09222, a humanized monoclonal antibody directed against the PACAP ligand, successfully inhibited PACAP38-induced cephalic vasodilation while reducing concomitant headache, so is thus a potential therapy against migraine [133].

When it comes to the CNS, PACAP38 is most abundant in the hypothalamus (likewise for CGRP), cerebral cortex, cerebellum, and brain stem [134]. Curiously, its receptors have a widespread distribution, some of them being particularly abundant in the cerebral cortex, amygdala, HIPP, thalamus, and hypothalamus [134,135]. Moreover, PACAP is directly involved in the regulation of monoamine synthesis and metabolism, brain-derived neurotrophic factor (BDNF) expression, and hypothalamic–pituitary–adrenal (HPA) axis activation [136], which, taken together with its distribution in anatomical areas that play a part in stress response and depression, allows the confirmation that this neuropeptide is closely related to the behavioral and endocrine responses to stress, as well as synaptic plasticity and neuroprotection [137].

Experimentally, the administration of PACAP in the paraventricular nucleus of the hypothalamus (PVN), the central nucleus of the amygdala (CeA), and the bed nucleus of the stria terminalis (BNST) has been shown to produce a stress-like response, and activate the HPA axis and the extrahypothalamic corticotropin-releasing factor (CRF) systems in rodents [138,139]. For example, PACAP-treated rats showed a dose-dependent increase in intracranial self-stimulation (ICSS), which is correlated with depressive-like behavior [138]; yet, this effect disappeared after administering a PACAP antagonist [140]. None of these depressive-like symptoms have been observed after the neuropeptide administration in PACAP knockout mice [141].

When it comes to patients with depressive disorders, little is known regarding the influence of PACAP on the onset, development, or clinical course of depression. Quantitative immunohistochemical staining of PACAP revealed elevated levels in the central BNST in postmortem samples of patients with MDD and bipolar disorder. However, this finding has only been observed in male subjects [142]. A significant positive correlation has also been reported between the Cornell depression score and PVN-PACAP-immunoreactivity in patients with Alzheimer’s disease and depressive or bipolar disorder [143].

Altogether, these data suggest that PACAP is involved in migraine and depression (Figure 2), indicating another molecular target that could be assessed to develop new pharmacological tools to enlarge the therapeutic armamentarium available to address both clinical entities.

#### 3.2.3. Neuropeptide Y

The craniocervical blood vessels, of great relevance in the etiopathology of migraine, are innervated by sympathetic fibers from the cervical and stellate ganglions [144,145] that store and release neuropeptide Y (NPY) [146,147,148,149]. This neuropeptide is crucial in controlling brain circulation due to its long-lasting vasoconstrictor properties [150,151].

Studies focusing on evaluating the NPY levels in plasma found an increase during attacks in migraine patients with aura and, to a lesser extent, in those without it [152]. In contrast, other authors found no variation during migraine attacks in this regard [153]. Similarly, in the CSF, some studies showed that NPY levels were higher in migraineurs [154], while other investigations found no alterations [155].

Recently, NPY signaling was shown to be involved in migraine via NPY receptor type 1 (Y1R). The microinjection of NPY into the medial habenula (MHb) exhibited analgesic and anxiolytic-like effects in the mouse model of glyceryl trinitrate (GTN)-induced migraine [156]. These effects are associated with the activation of Y1R, which, in turn, reduces the trigeminal activity evoked by the dura mater.

NPY also plays an essential role in mechanisms related to emotional reactivity and behavioral responses to stress [157], and its effects are influenced by the gut–brain axis [158]. In animal models of depression, decreased expression of NPY in the HIPP and hypothalamus [159] and reduced levels of NPY in the HIPP [160] have been observed. Furthermore, it was concluded that low NPY concentrations lead to depression, and certain antidepressants seem to increase NPY levels [161].

As has been demonstrated in migraine, the Y1R, NPY receptor 2 (Y2R), and NPY receptor 5 (Y5R) are potential therapeutic targets for neuroprotective and antidepressant drugs [162,163]. In animal models, the local injection of NPY into the medial prefrontal cortex induced antidepressant properties via Y2R treated with lipopolysaccharides [164]. In addition, Y1R agonists increase neuroblast growth, promoting BDNF release in the HIPP [165]. Intranasal administration of YR1 agonists also induces antidepressant-like effects by increasing BDNF [165].

In summary, these data point out the role of NPY in migraine and depression. Future studies with YR agonists will provide more information about their potential therapeutic usefulness.

#### 3.2.4. Substance P

Substance P (SP) is widely expressed in trigeminal sensory nerve fibers [166], primarily in the nucleus raphe magnus (NRM), locus coeruleus (LC), and periaqueductal grey (PAG) [167]. It plays a vital role in pain transmission [168,169] and vasodilation of the cerebral dura mater [170].

Several studies have focused on determining the alterations of SP in migraineurs. In spontaneous migraine attacks, no increase in cranial venous SP flow has been observed [153,171]. In contrast, there is an increase in salivary SP during spontaneous migraine attacks without aura [172]. Another study found higher levels of SP in platelets of migraine patients [173]. Interestingly, increased plasma SP concentrations during periods without headache have been detected in patients affected by episodic and chronic migraine [174,175].

Additionally, the evidence found to date indicates the involvement of the preferential receptor of SP, the NK1 receptor, in migraine. Antagonists of the NK1 receptor showed a robust effect in blocking plasma protein extravasation and decreasing the firing of second-order neurons in the trigeminal nucleus caudalis (TNC) [176]. Nevertheless, clinical studies with NK1 antagonists did not show a greater effect than the placebo in the acute management or as a prophylactic treatment of migraine [176,177,178,179]. Further studies are needed to investigate the role of NK1 receptors as potential new therapeutic targets for the treatment of migraine.

In the case of depressive disorders, the role of SP and NK1 receptors in the pathophysiology of depression is suggestive but not conclusive [180,181,182]. Opposite results were found when plasma and serum concentrations of SP were analyzed in patients with MDD. On one hand, it has been observed that plasma concentrations of SP are significantly reduced in patients with MDD [183]. However, additional studies found no differences in plasma SP levels between patients with MDD and healthy controls. No correlation has been found between plasma levels of SP and psychiatric symptoms or cognitive function [184]. In contrast, higher serum SP levels were identified in another study carried out in MDD patients [185].

In addition, SP is involved in the activation of the sympathetic system and the HPA axis in response to stressors [186]. In animal models, stress increases the release of SP in the AMY accompanied by anxious behavior [187]. Moreover, the central administration of SP elicited depressive and anxious behaviors in animals, whereas NK-1R antagonists induced anxiolytic and antidepressant-like effects [188,189,190].

Taken together, there is some evidence suggesting the involvement of SP in migraine, with more studies needed, especially regarding migraine–depression comorbidity.

#### 3.2.5. Orexins

There are two types of orexins (OX), orexin A (OXA) and orexin B (OXB) [191,192]. OXA binds to the OX1 (OX1R) and OX2 (OX2R) receptors, while OXB binds exclusively to OX2R [192,193,194]. OXA and OXB are synthesized in the lateral, posterior, and paraventricular nuclei of the hypothalamus [195,196,197,198], and their neurons project to nociceptive areas of the brain such as the LC, PAG and NRM, closely related to migraine [193,199,200,201]. OX1R is selectively expressed in the LC, while OX2R is expressed in the NRM [197,202].

There is evidence of the involvement of orexins in migraine. In migraine-related structures, neurons containing orexin showed increased activation during wakefulness and are inhibited during sleep [197]. Additionally, there is evidence that patients with chronic migraine have higher CSF levels of orexins [203]. Notably, the orexin system modulates the trigeminovascular system, inhibiting it through OX1R—which attenuates neurogenic dural vasodilation [204]—or activating it via OX2R. This has led to pharmacological studies evaluating the effects of drugs acting on these receptors.

Cady et al. demonstrated that dual OX1R and OX2R antagonists inhibited trigeminal sensory neuronal activation in rats [205]. On the other hand, the inactivation of OX1R in the basolateral amygdala (BLA) of rats increased photophobia, anxiety-like behavior, and social interaction deficits in the NTG-induced migraine model. Additionally, OX1R antagonism increased spontaneous migraine-like headache behaviors in NTG-treated rats. Interestingly, the blockade of OX1R failed to reduce hyperalgesia in this model, suggesting that OX1R play a more critical role in the modulation of emotional alterations rather than sensory processing in migraine [206].

However, in humans, the only clinical trial conducted to date, a double-masked, placebo-controlled study with an orexin receptor antagonist (filorexant), did not show its efficacy as a prophylactic treatment for migraine [207]. Further studies are needed to assess the therapeutic role of orexins in migraine. Based on the results obtained in animals, it might be interesting to investigate the role of pharmacological modulation of OXR1 in reducing anxiety and depression traits in migraine patients.

In the case of MDD, there is evidence supporting the involvement of orexins [208]; however, there is controversy as to whether orexin neurons are hyper- or hypoactive [209,210]. On the one hand, increased levels of orexins were detected in the hypothalamus of rats exposed to a model of depression induced via the neonatal administration of clomipramine [208]. On the other hand, previous studies showed that the size of soma cells and the number of orexin neurons are reduced in rodent models of depression [211,212,213]. In humans, clear changes in plasma orexin concentrations have been found, with MDD patients having lower levels than BP patients. Interestingly, depressed patients with suicidal ideation have higher levels, being proposed as a biomarker for preventing suicide [214].

Additional experiments sowed that orexins increased the firing frequency of dopaminergic neurons in the VTA [215], the brain area involved in depression [216,217]. In mice exposed to a chronic mild stress model, a paradigm of depression, the activity of orexin neurons projecting to the VTA was reduced. Furthermore, increasing orexin release in the VTA using optogenetics and chemogenetics significantly reversed depressive-like behaviors [218].

Concerning orexin receptors, the investigations carried out to date have indicated that OX1R and OX2R play distinct roles in anxiety and depression [213,219,220]. Firstly, studies on the orexigenic system in affective disorders have focused on the description of the anxiogenic and prodepressant actions of OrxA [221,222,223,224] through OX1R [220,225,226,227]. Subsequently, it was shown that OX2R activation, upon intracerebroventricular administration of an OX2R agonist, exhibited anxiolytic and antidepressant activity [228]. In a rat model of chronic mild stress, OX1R antagonism reduced depressive behaviors [229]. In the case of ORX2 antagonism, animal studies have provided conflicting results [230,231]. Interestingly, dual orexin receptor antagonists have shown antidepressant effects [224,232].

In humans, dual orexin receptor antagonists, suvorexant and lemborexant, have been approved for the treatment of insomnia disorder in the United States, characterized by difficulties with sleep onset and/or sleep maintenance in adults based on the results of pivotal phase 3 studies. In the case of MDD, a clinical trial found that suvorexant, a potent and highly selective antagonist of OX2R, significantly reduced depressive symptoms and alleviated insomnia in MDD patients with sleep disturbances [233].

In short, orexine receptors appear to be promising targets for treating anxiety and depression in migraine patients, deserving further exploration.

### 3.3. Sexual Hormones

#### 3.3.1. Estrogens

As previously mentioned, migraine presents differences in prevalence based on sex, with the number of cases being three times higher in women [1]. One factor that may explain the sex disparity is the levels of ovarian steroid hormones and how these vary according to the menstrual cycle. It is worth noting that there is an increase in the prevalence of migraine at puberty, with a greater incidence in women of childbearing age, affecting 24% of women between 30 and 39 years of age [234,235]. Among all cases of women with migraine, approximately 22% of the total were menstrual migraines in women of childbearing age. Migraines without aura were more frequent than migraines with aura, with menstrual-related migraine (MRM) being more common than pure menstrual migraine (PMM) [236]. Moreover, attacks caused by menstrual migraine are more severe, painful, disabling, last longer, and usually course with nausea and allodynia [237,238,239].

In the case of menstrual migraines, the hypothesis of estrogen withdrawal has been proposed as the etiological mechanism. Accordingly, the precipitation of migraine headaches is related to a drop in estrogen levels below 40–50 pg/mL [240]. Aspects such as the “magnitude of the decline” and the “residual threshold” have been proposed as precipitating factors. The first one assumes that a minimal reduction in estrogen is needed to trigger a migraine attack. The second one assumes that a minimal concentration of estrogen in the blood must be maintained to prevent migraine [241]. Another aspect is the rate of estrogen decrease. In the Study of Women’s Health Across the Nation (SWAN) Daily Hormone, women with migraine had a more rapid premenstrual decline in estrogen levels than controls [242].

Several mechanisms have been proposed to underlie the association between decreased estrogen levels and migraine. Estrogen receptors are highly expressed in brain regions involved in pain processing, such as the thalamus, PAG, AMY, and trigeminovascular system [243]. Estrogen depletion increases the susceptibility of vessels to prostaglandins, activating the endothelial cell nitric oxide synthetase, and, consequently, the production of the vasodilator NO [244]. Additionally, estrogen depletion reduces endogenous opioid activity, as well as having effects through the serotonergic and dopaminergic systems [245]. Estrogens increase the expression of the rate-limiting enzyme tryptophan hydroxylase and reduce the serotonin reuptake [245,246,247]. Thus, estrogen depletion impacts neuronal excitation and pain perception, increases allodynia, induces central nervous system sensitization, and promotes cortical-spreading depression [248,249,250].

As in the case of migraine, sex is also a risk factor for depression. MDD is more common in women than men, a difference that persists into old age [251,252]. Symptoms are generally more severe in women; feelings of loneliness and low self-perception of health are common among depressed women, experiencing prolonged or recurrent depression more than depressed men, with a younger onset and lower quality of life [253,254,255,256]. Notably, there are pieces of evidence supporting the impact of fluctuations in the ovarian estrogen hormone levels on women’s well-being [257,258]. There is a relationship between the menopausal period and the onset of depressive disorders. Moreover, the hormonal fluctuations before menstruation and during pregnancy, the puerperium, or the perimenopausal period are closely linked with mood disorders [257]. Additionally, sex also impacts antidepressant efficacy [259]. For instance, postmenopausal women have a diminished response to antidepressants compared with younger women.

As in migraine, there is a relationship between low sex hormone levels and increased prevalence of depression [260]. Furthermore, the administration of exogenous estrogens has antidepressant effects in depressed women, with the effect more significant if administered during perimenopause, in the form of a transdermal patch, or the postpartum period [261,262,263]. However, there is controversy, as other studies have shown that estrogens did not improve mood in postmenopausal women or even increase the risk of cognitive impairment and stroke [264,265,266].

Estrogens also play an important role in the efficacy of antidepressants by regulating the CNS [267]. As mentioned before, estrogens are linked with the serotoninergic system. Estrogens participate in the synthesis and degradation of 5-HT, in the density of serotonergic receptors, and in the expression of 5-HT-related genes [268]. Notably, estrogens are thought to promote serotonergic signaling to exert antidepressant effects [269,270,271].

Additionally, estrogens activate transcription factor pathways, changing the gene expression of trophic factors such as BDNF [272], which is involved in neuronal survival and differentiation, synaptic transmission, learning, and memory [273,274,275,276]. BDNF has been extensively studied and linked to depression [277,278,279,280,281]. Studies have shown that estradiol and BDNF activate similar signaling pathways. Moreover, estradiol increases BDNF expression and proteins, which fluctuates following the same dynamics of change as estradiol during menstruation [282,283,284]. In addition, the exogenous administration of estradiol to ovariectomized animals has been shown to reverse BDNF depletion and prevent the development of depressive behavior [285,286,287].

In summary, although further studies are needed, all this cumulative evidence supports the involvement of estrogens in migraine. Fluctuations in estrogens are essential in the pathophysiology of migraine. Moreover, changes in estrogens also influence mood state and directly influence the brain circuits involved in emotional regulation. The clinical implications of these findings suggest that estrogen status may be important for migraine and depression comorbidity and that estrogen treatment or replacement may deserve further exploration as a pharmacological option for women (Figure 3).

#### 3.3.2. Progesterone

Progesterone is a neurosteroid hormone produced by the ovaries and placenta in women and by the adrenal glands and brain in both sexes. In the CNS, progesterone is synthesized by glial cells and neurons [289,290]. Its action on the progesterone receptor mediates the physiological effects of progesterone. Beyond its effects on controlling reproduction, progesterone has been found to play a role in developing and maintaining the neurons in the brain [291,292,293]. In the case of migraine, progesterone receptors are involved in regulating pain sensitivity and migraine susceptibility in women [294].

Cumulative evidence highlights the potential for progesterone to modulate sensory neurotransmission and vascular responses in a complex manner, indicating that progesterone primarily serves as a modulator rather than as an elicitor. Progesterone downregulates estrogen receptors, which in turn reduces the activation of trigeminovascular pain pathways [295]. Additional progesterone actions observed include the control of vasodilatation by decreasing histamine secretion from mast cells and prostaglandin production [246,296,297].

Progesterone also plays a major role in depression, possibly mediated by its neurosteroid derivative tetrahydroprogesterone [298]. Peripheral progesterone has a protective effect on mood and has therefore been tested for the treatment of postpartum depression [299,300]. In ovariectomized mice exposed to a model of depression, progesterone administration was shown to reduce anxiety and depression via changes in the gut microbiota [301]. These findings suggest that progesterone may be useful in improving depressive symptoms in menopausal women [302,303].

Thus, the broad role of progesterone in migraine and the efficacy of progesterone modulating depressive symptoms make it mandatory to further explore its potential therapeutic role in migraine–depression comorbidity (Figure 3).

#### 3.3.3. Prolactin

Prolactin (PRL) is another crucial hormone involved in migraine. PRL receptors are expressed in the neurons of the trigeminal ganglia as well as in dural afferent neuronal fibers, which are structures involved in the nociception and pathogenesis of migraine [304,305].

Clinical and preclinical studies have supported the involvement of PRL and its receptors in migraine, with important sex differences. The dural administration of prolactin-induced long-lasting migraine-like behaviors only in women [306]. Moreover, higher expression of PRLP receptors has been observed in women compared to men in trigeminal ganglion sensory neurons and in the neuronal fibers that innervate the dura mater [306,307,308,309]. Importantly, PRL is also associated with CGRP, the serotonin system, and PACAP-38 [130,306,308]. Evidence has indicated that PRL is involved in the modulation of neuronal excitability and pain mainly via its action on TRPV receptors [310,311,312].

Alterations in PRL have been observed in migraine patients, being associated with the progression of migraine. Thus, increased PRL levels are considered as a worsening factor for migraine [313,314]. Preventive drugs and triptans also modified PRL levels [315,316]. There is evidence of the reduction in PRL levels after the administration of triptans, such as sumatriptan [316].

Prolactin has also been studied in depressive disorder. The hypothalamic–prolactin axis is dysregulated in depressive patients with suicidal behavior, especially if the suicide attempt has been severe [317]. It was suggested that prolactin can reduce stress by attenuating the responsiveness of the HPA axis [318]. However, other studies showed that prolactin secretion is increased in stressful situations [319]. Notably, it was observed that plasma prolactin levels were higher in individuals with MDD than in controls, suggesting that prolactin dysregulation may be a feature of MDD [320].

Altogether, all these data indicate PRL and their receptors as new candidates to be further explored in the migraine–depression comorbidity (Figure 3).

#### 3.3.4. Oxytocin

Oxytocin (OT) is highly implicated in the process of migraine initiation and may, in turn, be a potential therapeutic target [321]. OT is a pleiotropic hypothalamic neurotransmitter that exerts an antinociceptive effect via its OTR receptor to inhibit trigeminal neuronal excitability. This is of relevance because of the involvement of the hypothalamus in the different phases of migraine, increasing blood flow in this brain region during premonitory symptoms and processing nociception [230,322]. Activation of the OTR results in intracellular Ca2+ mobilization, inhibiting nociception through GABAergic signaling, inhibition of transient potassium current, desensitization of TRPV1 channels, and disruption of NMDA-coordinated neuronal network activity [323,324,325].

Additionally, spinal oxytocin reduced the neuronal firing of the trigeminocervical complex caused by meningeal electrical stimulation in rats [325]. Additional studies showed that OTR are expressed in the trigeminal vascular system in rats but not in the cranial arteries [326,327]. OT receptors are present in the neuronal structures closely related to migraine precipitation, such as the trigeminal ganglion and caudal trigeminal nucleus.

In an animal model of chronic migraine, the intranasal administration of OT abolished central sensitization by regulating synaptic plasticity [328]. Moreover, in menstrual migraine, alterations in OT levels and OTR expression appear to be involved in the activation of meningeal trigeminal nociceptors and the subsequent risk of migraine attacks during menstruation [329].

Oxytocin is also crucial in modulating emotional behaviors, with evidence of anxiolytic and antidepressant effects [330,331,332,333,334]. Among other aspects, oxytocin has been associated with attachment, social bonding, feelings of trust, positive communication, altruism, and empathy in different studies [335,336,337,338,339,340]. Oxytocin is an essential neuromodulator in the amygdala, hypothalamus, and nucleus accumbens, brain regions closely related to depression [341,342].

Exogenous oxytocin has been shown to significantly decrease anxious and depressive behaviors in mice, with its effects influenced by sex, estrous cycle, and hormone levels [343]. The effect of exogenous oxytocin administration is attenuated in females, as shown in different animal models, involving social defeat, social avoidance, and social preference and avoidance induced by maternal defeat [344,345,346].

On the other hand, oxytocin may affect neuronal plasticity in response to environmental conditions, thereby modifying behavioral and psychological outcomes [347,348]. The relationship between oxytocin and depression is not fully understood, as some studies did not report consistent findings [349,350,351].

Concerning the therapeutic potential of the oxytocinergic system in depressive disorders, one clinical trial showed an improvement in mood when oxytocin was administered for two weeks together with escitalopram treatment [352]. Notably, after the use of different antidepressant treatments in depressive patients, serum oxytocin levels were not affected, despite reduced depressive scores [353]. However, this does not imply that oxytocin is unrelated to the development of depression or that it is not useful as a prophylactic [354]. One promising finding is that oxytocin may be used as an adjunct drug to antidepressant treatment or to treat specific aspects of depressive disorders [355].

Therefore, the oxytocinergic system is an ideal new potential therapeutic target deserving further exploration in the treatment of depression in migraine (Figure 3).

### 3.4. Glutamate and GABA

Glutamic acid/glutamate and γ-aminobutiric acid (GABA) are the main excitatory and inhibitory neurotransmitters in the CNS, respectively. Glutamate is taken up from the circulation or directly synthesized from glutamine, α-ketoglutarate, and 5-oxoproline and catabolized in the neurons and glia [356]. GABA’s precursors are mainly glutamate, pyruvate, and other amino acids [357]. Dysregulations in both glutamatergic and GABAergic neurotransmission systems have been continuously linked to different neuropsychiatric disorders, including migraine and depression.

It is known that migraine pain-relay centers, including the trigeminal ganglion, trigeminocervical complex (TCC), and sensory thalamus, contain glutamate-positive neurons [358]. However, the brains of migraineurs differ pharmacologically from those of nonmigraine sufferers, and it seems that glutamate may play a major role in such differences [359]. For example, magnetic resonance spectroscopy studies of the cortex and the thalamus found higher interictal glutamate levels in the visual cortex and thalamus of migraine patients but no group differences in GABA levels, supporting the hypothesis of cortical and thalamic hyperexcitability in migraine driven by the excess availability of glutamate [360,361]. Despite these results, a recent study also found increased GABA levels in the visual cortex from interictal toward the preictal state for migraine patients compared with healthy controls, thus supporting a potential role for GABA in migraine [362].

Furthermore, CSF concentrations of glutamate have also been found to be higher in migraineurs than in healthy volunteers, confirming the excess of neuroexcitatory transmission in the CNS [358]. This glutamate release affects the spinal dorsal horn, causing glutamate receptor activations and central sensitization, with allodynia and hypersensitivity being the most common clinical consequences of this excitatory disbalance, suggesting a defective cellular reuptake mechanism for glutamate in migraine patients at the neuronal/glial level [363]. More specifically, multiple pain models suggest that this central sensitization includes multiple mechanisms of synaptic plasticity caused by changes in the density, nature, and properties of ionotropic and metabotropic glutamate receptors [364,365]. Even other characteristic features such as aura seem to be directly or indirectly caused by this hyperexcitatory status, since the local release of glutamate by neurons is thought to initiate the cortical-spreading depression that causes this visual symptom [366].

Additionally, glutamate is involved in the nociception of migraine through its kainate receptors. This observation is based on studies that showed how preventive, approved treatments such as topiramate inhibit third-order neurons responding to trigeminovascular stimulation and to selectively block the excitation induced by kainate receptor agonists but not by N-metil-D-aspartate (NMDA) or α-amino-3-hydroxy-5-methyl-4-isoxazolepropionic acid (AMPA) receptor agonists [367]. Nevertheless, noncompetitive NMDA receptor channel blockers like memantine have demonstrated significant effects in reducing headache frequency and mean disability scores when given as a preventive treatment of refractory migraine [368,369,370]. Other anticonvulsants, such as lamotrigine, have also exerted potential prophylactic properties that could be superior to topiramate. However, the outcomes in this regard have been conflicting [371,372].

The glutamate signaling pathway is also indirectly modulated by other approved treatments for migraine, such as triptans and CGRP monoclonal antibodies. Triptans have been shown to interfere with the release of glutamate from the primary afferents in the TCC by decreasing the amplitude of glutamatergic excitatory postsynaptic currents and reducing the frequency of spontaneous excitatory postsynaptic currents. These actions are potentially mediated by the presence of 5-HT_1D_ and/or 5-HT_1B_ receptors on the presynaptic terminal, thus affecting presynaptic Ca^2+^ influx [373]. On the other hand, anti-GCRP monoclonal antibodies are thought to prevent NMDA- and AMPA-evoked firing potentiation and the nociceptive activation of second-order neurons [374].

Regarding depression, the glutamate hypothesis was proposed in the 1990s, when antagonists of the NMDA receptor were found to possess antidepressant-like mechanism [375]. To date, many clinical and animal studies have reported impairment of the glutamatergic system in various limbic and cortical areas of the brain of depressed subjects [376]. Several authors have also reported the decreased expression of NMDA [377] and AMPA, alongside the decreased availability of metabotropic receptor 5 (mGluR5) in the PFC, cingulate cortex, thalamus, and hippocampus in depressed individuals [378,379].

The crosstalk between the glutamatergic and serotonergic systems is essential to understand the antidepressant effect of drugs [380]. mGluR2 and -3 antagonism exert antidepressant effects in rodent models similar to those of ketamine, with shared synaptic response and neural mechanisms, implicating the serotonergic system [380]. This blockade increases the extracellular5-HT levels in the rat medial prefrontal cortex (mPFC) through activation of the AMPA receptor (which leads to an increase in the activity of 5-HT neurons in the dorsal raphe nucleus (DRN), presumably via the mPFC-DRN projection), activating downstream synaptogenic signaling pathways (e.g., BDNF, mTOR) [381]. The antidepressant actions of ketamine are blocked under the pharmacological depletion of 5-HT in the brain [382]. This observation shows how crucial serotonin–glutamate interaction is to understand the pathophysiology of MDD and develop therapeutic tools that can modulate this crosstalk.

The role of GABA in depression has also been extensively studied. GABAergic interneurons are identified by their expression of specific receptors: somatostatin (SST), parvalbumin (PV), and serotonin 3A (5-HT_3A_). SST and PV interneurons make up 30 and 40%, respectively, of the total GABAergic neuronal pool [383]. Postmortem studies of depressed patients identified reduced levels of SST and PV interneurons in PFC as well as in other cortical areas [384]. Additionally, diminished levels of SST messenger RNA (mRNA) have been found in several brain regions in depressive patients, including the dorsolateral PFC [385] and amygdala [386], key regions involved in emotional processing. Furthermore, reduced expression of MDD subjects [386,387]; treatment with various antidepressants, electroconvulsive therapy (ECT), and cognitive-behavioral therapy tends to restore GABA levels [388,389].

Taken together, the evidence presented above suggests that by targeting the glutamatergic system through the NMDA, AMPA or mGLU receptors, an impact could be made on patients who suffer from migraine or MDD concomitantly. Also, the modulation of the GABAergic pathway could ameliorate symptoms in patients who suffer from these two entities; yet, this would require a deeper understanding of how inter-related migraine and MDD are regarding GABA signaling.

### 3.5. Immune System

The accumulated evidence clearly suggests that the activation of the immune system leading to peripheral and central inflammatory phenomena is a common element in the pathomechanism of migraine and depression. In this sense, exposure to stressful stimuli might be a shared environmental factor in both diseases. Stress has been recognized as one of the most critical factors exacerbating migraine pain. It is responsible for meningeal vasodilation and an increase in vascular permeability, favoring the influx of inflammatory cells that would ultimately lead to the activation of microglia, resulting in a neuroinflammatory process. Similarly, in depression, exposure to chronic stress is commonly linked to chronic inflammation, with a pronounced increase in proinflammatory cytokines crossing the blood–brain barrier (BBB), which, in turn, activates the microglia and induces a neuroinflammatory state [390]. Hence, this section aims to recapitulate the main findings about the inflammatory processes shared by migraine and depression.

The pathophysiological basis of migraine is based on the local terminal release of trigeminovascular afferents products that are able to provoke not only the dilation of meningeal vessels but also a very evident neuroinflammatory state [391,392]. It has been proposed that cortical-spreading depression (CSD), an electrophysiological phenomena underlying migraine aura [393], as well as stress and/or hormonal fluctuations could underlie two independent cascades generating an inflammatory state in the intracranial meninges mediated to a large extent by dural immune cells [394] and microglia [395], which sensitize and activate trigeminal meningeal nociceptors, also referred as neurogenic inflammation [396,397]. Additionally, a dysfunction of the glymphatic system has been recently proposed as another mechanism involved in meningeal inflammation and trigeminal nociception [398,399,400].

Overall, this neuroinflammatory state leads to vasodilation, plasma extravasation secondary to capillary leakage, edema, and mast-cell degranulation. Mast cells have a pivotal role in the neuroinflammatory state associated with migraine [401,402,403], which was first introduced by Sicuteri et al. in the 1950s [404], due to their close proximity to both vasculatures and nerve fibers [403]. Mast cells can selectively release proinflammatory cytokines such as tumor necrosis factor-alpha (TNF-α), IL-1β, and IL-6, as well as lipid-derived mediators (i.e., leukotrienes and prostanoids), without requiring the degranulation phenomenon [405,406].

Alterations in several major proinflammatory cytokines, including TNF-α, IL-1β, IL-6, and IL-8, have been found in different biological samples of migraineurs [80]. In fact, TNF-α presence is commonly increased in the plasma, serum, and/or urine during migraine attacks and attack-free intervals, suggesting the pathogenic role of TNF-α in the inflammatory state of these patients [407,408,409,410,411,412]. Likewise, circulating IL-1β, IL-6, or IL-8 levels are often enhanced in migraineurs compared to healthy controls, especially during the ictal stage of migraine [413,414,415,416], as recently shown by a meta-analysis [417].

In addition to the role of cytokines in migraine, there is a growing interest in studying changes in the immune system at the cellular level. These may reflect specific alterations associated with the pathophysiology and severity of migraine. Among the most relevant findings that can be highlighted are a reduction in CD4+ CD25+ regulatory T cells [418,419,420]; higher CD3, CD4, CD8, and CD19 in patients with chronic migraine compared with episodic migraine [421]; or increased proportion of Treg CD45R0+ CD62L−, and CD45R0-CD62L− cells [422]. Also, a study comparing episodic migraine patients with healthy controls revealed higher values of several lymphocyte-related blood parameters in migraineurs. Interestingly, a lower value of CD4+ T_EM_ (effector memory helper T lymphocytes) appears to represent a potential biomarker determining the severity of migraine [423]. In line with these results, a recent report showed a significantly lower percentage of blood CD3+ CD4+ helper T cells and CD4+ CD25+ regulatory T cells in migraineurs, suggesting a dysregulated peripheral immune cell homeostasis [424].

Another approach used to assess the pathophysiological role of the immune system in migraine is to evaluate inflammatory response markers by obtaining the ratios between different cellular subtypes. Serum neutrophil/lymphocyte ratio (NLR), monocyte/lymphocyte ratio (MLR), and platelet/lymphocyte ratio (PLR) may represent biomarkers associated with different migraine clinical features such as the attack period, aura, or family history [425].

Genome-wide gene expression studies have been performed to characterize further immune function alterations associated with migraine. In this sense, whole-blood next-generation RNA sequencing revealed differentially expressed genes related to immune and inflammatory pathways, including those expressed in microglial cells [426]. Furthermore, RNA sequencing in peripheral blood mononuclear cells (PBMCs) was carried out to analyze the transcriptome of migraineurs during and between attacks. This study suggested the importance of inflammatory pathways and the potential contribution of several cytokines to migraine susceptibility, upregulated in both interictal and ictal samples from migraineurs compared to healthy controls [427]. Moreover, small-RNA sequencing analyses in PBMCs followed by mRNA transcriptomics unveiled that several micro-RNAs (miRNAs) related to immune and inflammatory responses, neuroinflammation, and oxidative stress were differentially expressed during and between headaches in migraineurs compared to healthy volunteers [428]. A recent study identified 45 shared genes between migraine and MDD via single-cell RNA sequencing. Among those related to inflammation, IL-1β was highly expressed in microglia cells [429].

The latest evidence also points to the gut–brain–immune axis as a critical player in migraine’s etiology, pathogenesis, frequency, and severity. However, more research is needed to elucidate and understand the underlying mechanisms. For now, it is hypothesized that alterations in the microbiome may impact the levels of certain neurotransmitters, mainly serotonin, as well as on the peripheral inflammatory state and its reflection in the central nervous system [430,431].

With regard to depression, the accumulated preclinical and clinical evidence clearly indicates that there is systemic immune activation, which is reflected in significant changes in the levels of inflammatory markers and in the type and number of immune cells [432,433,434]. In fact, increased levels of peripheral proinflammatory cytokines, mainly IL-1β, IL-6, and TNFα, have been described in depressed patients [435,436]. Likewise, several recent studies have shown that alterations occur in the cellular component of the immune system related to depression, with an increase in circulating monocytes [437,438], Th17 cells, Th17:Treg ratio [439], and Treg cells [440,441], as well as an association with NLR and PLR [442]. It is important to note that differential changes associated with an impaired peripheral inflammatory state have been described according to symptomatology [443], including suicidal risk [444,445], as well as the severity [446,447], clinical progress [448,449], or treatment response [450,451,452,453] of the depressive disorder, among other relevant aspects [454].

Peripheral molecular and cellular immune dysfunction affects the CNS, inducing a neuroinflammatory state that is becoming better understood in the context of depression [455]. Among the different neuropathological findings that have been described, the activation of microglia seems to be one of the most relevant phenomena [456] and has been proposed to be related to alterations occurring in the hypothalamus–pituitary–axis (HPA) axis as a consequence of stress exposure [457], the latter being one of the shared etiological events between depression and migraine, as discussed above. Interestingly, a very recent study identified 45 shared genes between MDD and migraine via single-cell RNA sequencing. Among those related to inflammation, IL-1β was highly expressed in CNS microglia cells [429].

Finally, and briefly, the gut–brain–immune axis appears to play a very relevant role in the pathogenesis of depression, as also discussed for migraine. There is growing evidence linking alterations in the gut microbiota of depressed patients with their impact on the immune system and the eventual reflex on the CNS, being related with disturbances in monoamine neurotransmission and neuroinflammation [458,459,460,461,462,463].

In conclusion, functional changes, either molecular or cellular, in the peripheral and central immune system appear to be a critical pathophysiological feature that may be shared, at least in part, between migraine and depression. This clearly justifies the need for further research to better understand the common mechanisms involved and how they might help to improve the diagnosis, follow-up, or treatment of patients with this comorbidity.

## 4. Discussion

In summary, the present review provides relevant information about the common pathways that are altered in migraine–depression comorbidity (Figure 4). The pathophysiology of migraine is complex and is not yet fully understood. Mainly, neurotransmitters and peptides closely related to the control of pain at the peripheral and central levels have been proposed to play a relevant role in different pathophysiological aspects of migraine. Serotonin is highly relevant in migraine since the levels of 5-HT are considered an indicator of pain-modulating system dysfunction. Moreover, genetic studies revealed that two polymorphisms, VNTR STin2 and 5-HTTLPR, are associated with migraine. More importantly, GWAS studies supported the finding that migraine is a polygenetic disorder. In the case of depression, vast information supports the role of serotonin. However, studies focused on examining the role of serotonin in the comorbidity of depression and migraine are scarce. Despite this, the findings are promising, since a more pronounced reduction in brain volume has been described in migraine patients with depression, as well as alterations in the activity of different brain regions, such as the medial prefrontal cortex, dorsal raphe, and thalamus.

CGRP is closely related to migraine, being involved in vasodilatation and central and peripheral sensitization, contributing to pain exacerbation. The role of CGRP in migraine is the rationale behind the use of anti-CGRP monoclonal antibodies in episodic/chronic migraine that is unresponsive to other preventive therapies. Interestingly, some evidence shows alterations in CGRP in patients with MDD; curiously, preliminary studies showed promising results, with anti-CGRP monoclonal antibodies improving depression in individuals with migraine.

In addition, PACAP is closely related to migraine, controlling vasodilatation, modulation of the parasympathetic nervous system, mast-cell degranulation, and activation of the trigeminovascular system. Evidence points to the potential utility of monoclonal antibodies for PACP38 in migraine. Regarding PACAP in MDD, some published work shows elevated levels in the central BNST in postmortem samples of male patients with MDD; yet, there is no available evidence on how PACAP could impact patients with migraine–MDD comorbidity.

Data also support the involvement of NPY in migraine and depression, showing that the pharmacological modulation of NPY receptors, such as Y2R and Y5R, displays antidepressant effects. In the case of SP and its preferential receptor NK1, which participates in the control of pain transmission and vasodilatation of the cerebral dura mater, only one clinical study was conducted to date to analyze the efficacy of NK1 antagonism in migraine, reporting negative results. Interestingly, NK1 antagonism induced anxiolytic and antidepressant-like effects in animal models. Likewise, the pharmacological modulation of orexin receptors also represents a potential new avenue to explore in the treatment of migraine and depression.

Sexual hormones, mainly estrogens, are of relevance for migraine and depression. Fluctuations in sexual hormones impact women’s well-being. There is a close relation between estrogens and the efficacy of antidepressants since estrogens are thought to promote serotoninergic signaling and are linked with neuroplasticity. Moreover, progesterone is a key modulator of sensory neurotransmission and vascular responses that has emerged as an important factor, especially in women with migraine and depression. Last, but not least, prolactin is closely related to CGRP, serotonin, and PACAP. It is also a key player in the differences between men and women in migraine. Finally, oxytocin has been strongly implicated in the process of migraine initiation and emotional disturbances, and may, in turn, be a therapeutic target for migraine-and-depression comorbidity.

Alterations in the excitatory/inhibitory balance have been described in depression. Additionally, cumulative evidence also supports alterations in glutamate and, to a lesser extent, GABA in migraine. The evidence suggests that targeting glutamate receptors, for example, NMDA or AMPA receptors, may be of interest for those patients who suffer migraine and MDD concomitantly.

Finally, neuroinflammation is also of relevance in psychiatric disorders such as depression; more recently, it has been explored as an etiopathological mechanism of migraine, with promising results.

Although there is evidence for the role of neuropeptides, sex hormones, glutamate/GABA, and the immune system for both clinical entities separately, there are no studies that have investigated their role in depression–migraine comorbidity. Considering the clinical relevance of this comorbidity, further studies are needed that will focus on examining the role of these promising systems, hormones, or peptides in migraine patients with MDD.
Figure 4The etiopathogenic mechanisms involved in migraine and depression. GLU: glutamate, GABA: gamma-aminobutyric acid, 5-HT: 5-ydroxytryptamine, DRN: dorsal raphe nucleus, TNC: trigeminal nuclear complex, NPY: neuropeptide Y, PACAP: pituitary adenylate cyclase-activating polypeptide, CGRP: calcitonin -gene-related peptide, SP: substance P.
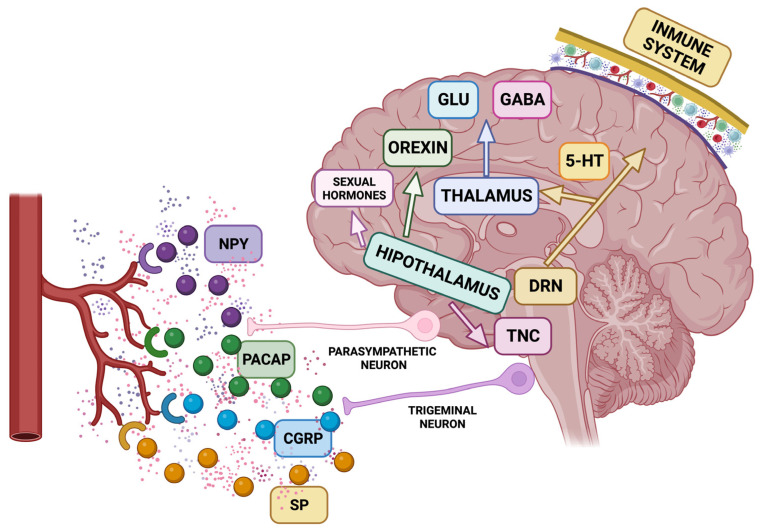


## Figures and Tables

**Figure 2 biomolecules-14-00163-f002:**
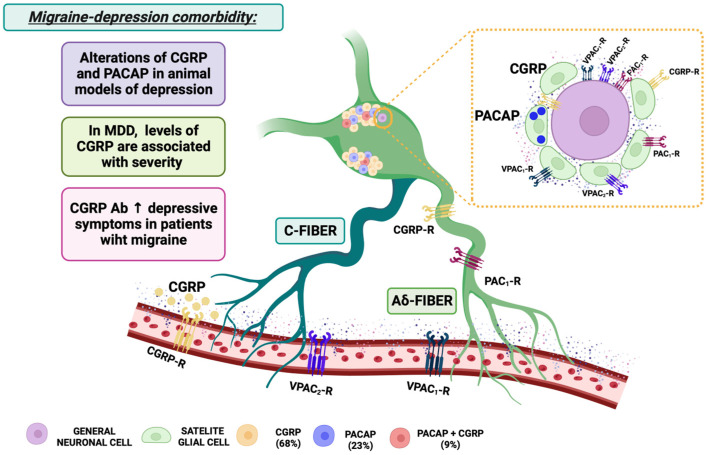
The involvement of the CGRP and PACAP in migraine and depression. CGRP: calcitonin-gene-related peptide; PACAP: pituitary adenylate cyclase-activating polypeptide; MDD: major depressive disorder; Ab: monoclonal antibody; VPAC1-R: vasoactive intestinal polypeptide receptor 1; VPAC2-R: vasoactive intestinal polypeptide receptor 2; PAC1-R: pituitary adenylate cyclase-activating polypeptide type I receptor; CGRP-R: calcitonin-gene-related peptide. Figure adapted from [112].

**Figure 3 biomolecules-14-00163-f003:**
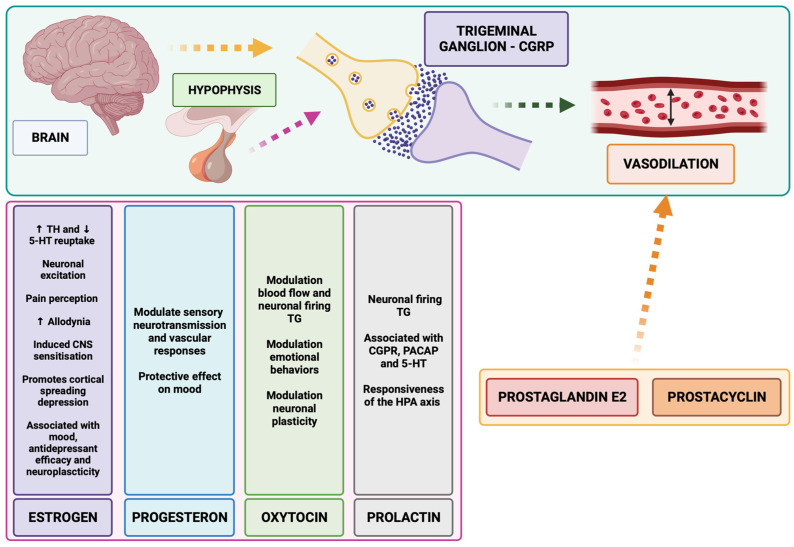
The role of sexual hormones in migraine and depression. TH: tyrosine hydroxylase; 5-HT: serotonin; CNS: central nervous system; TG: trigeminal ganglion; CGRP: calcitonin-gene-related peptide; PACAP: pituitary adenylate cyclase-activating polypeptide; HPA: hypothalamic–pituitary–adrenal axis; ↑ increase; ↓ decrease. Figure adapted from [288].

## Data Availability

Not applicable.

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
