# Peer review of "Understanding the Biological Relationship between Migraine and Depression"

_biomolecules, 2024, doi:10.3390/biom14020163_

Round 1

Reviewer 1 Report

Comments and Suggestions for Authors

The authors are trying to explain the relationship between migraine and depression from the perspective of various neuropeptides such as CGRP, PACAP, estrogen, etc.

These findings are very important for scientifically elucidating migraine and depression.

If we dare to say more, it’s better to make the figures (cartoons) in the text easier to understand. Instead of forcing them into one figure, it would be nice to have multiple figures (cartoons).

Author Response

We would like to thank the reviewer for his/her comments. We have added three new figures on the role of serotonin, PACAP and CGRP peptides and sex hormones. 

Reviewer 2 Report

Comments and Suggestions for Authors

The isues related to the coexistence of migraine and mood disoredrs and depression in particular have been a subject of clinical research for a long time. The epidemiological evidence of the association of the two disorders is strong and the evidence of shared biochemical characteristics is emerging though difficult to interpret. This review serves an important objective in looking at each candidate compound in details and assess the evidence of their contribution to pathogenesis of migraine and their roles in mood disorders.

The authors have made a comprehensive review of the literature and made appropriate and careful analysis of the findings. The authors were careful in presenting a balanced analysis with focus on the scientific evidence and reserved the opinions to an appropriate level withiut overstating or underplaying the value of each piece of evidence.

Although there are no strong conclusions that can be drawn and the issue continues to be an important area for future research, nevertheless this review will allow better understanding of the association of the two disorders and may help in highlighting the need for future research.

I spotted two typos that need corections:

Page 2, line 63: remove "and" at the end of the sentence

Page 3, Line 145: gene Not gen    

Author Response

We greatly appreciated the comments made by the reviewer. We have fixed the typo errors mentioned by the reviewer.